# Difference between the Effects of Peripheral Sensory Nerve Electrical Stimulation on the Excitability of the Primary Motor Cortex: Examination of the Combinations of Stimulus Frequency and Duration

**DOI:** 10.3390/brainsci12121637

**Published:** 2022-11-29

**Authors:** Masaaki Sato, Hitoshi Mutai, Jun Iwanami, Anna Noji, Sayaka Sugimoto, Kana Ozawa, Akira Sagari

**Affiliations:** 1Department of Occupational Therapy, School of Health Sciences, Faculty of Medicine, Shinshu University, Matsumoto 390-8621, Japan; 2Department of Rehabilitation, Aizawa Hospital, Matsumoto 390-8510, Japan; 3Department of Rehabilitation, Toyama Rosai Hospital, Uozu 937-0042, Japan; 4Department of Rehabilitation, Suwa Kyoritsu Hospital, Okaya 393-0077, Japan

**Keywords:** electrical stimulation, stimulation paradigm, motor-evoked potential, transcranial magnetic stimulation, design of experiments

## Abstract

Peripheral sensory nerve electrical stimulation (PES) excites the primary motor cortex and is expected to improve motor dysfunction post-stroke. However, previous studies have reported a variety of stimulus frequencies and stimulus duration settings, and the effects of these different combinations on primary motor cortex excitability are not clear. We aimed to clarify the effects of different combinations of stimulus frequency and stimulus duration of PES on the excitation of primary motor cortex. Twenty-one healthy individuals (aged > 18 years, right-handed, and without a history of neurological or orthopedic disorders) were included. Each participant experienced three different stimulation frequencies (1, 10 and 50 Hz) and durations (20, 40 and 60 min). Motor-evoked potentials (MEPs) were recorded pre- and post-PES. The outcome measure was the change in primary motor cortex excitability using the MEP ratio. We used a D-optimal design of experiments and response surface analysis to define the optimal combination within nine different settings inducing more satisfying responses. The combination of stimulation frequency and stimulation time that maximized the desirability value was 10 Hz and 40 min, respectively. The results of this study may provide fundamental data for more minimally invasive and effective implementation of PES in patients with stroke.

## 1. Introduction

Electrical sensory input has been reported to contribute to the improvement of motor dysfunction in patients with stroke and has been widely used in rehabilitation medicine in recent years [1]. Peripheral sensory nerve electrical stimulation (PES) can be applied using low-intensity electrical stimulation without patients experiencing fatigue or pain. Prolonged PES to the ulnar nerve can increase the excitability of the primary motor cortex in healthy participants [2,3]. One of the mechanisms by which PES increases the excitability of the primary motor cortex is peripheral sensory nerve stimulation as an ascending input, leading to the de-inhibition of the gamma-aminobutyric acid inhibitory system, which in turn leads to activation of the primary sensory cortex in the area of body part reproduction, as well as activation of the primary motor cortex [3,4]. Activation of the primary motor cortex by PES is thought to be effective in improving motor paralysis in stroke patients, and the effects of interventions combining various rehabilitation approaches and PES have recently been reported [5,6,7]. The combination of PES and task-oriented training improved upper limb function in patients with subacute [5] and chronic [6] stroke. A study also revealed that low-frequency repetitive transcranial magnetic stimulation (TMS), followed by PES and occupational therapy, was effective in improving upper limb function in patients with chronic stroke [7].

However, a systematic review [8] evaluating the effects of PES on cortical plasticity showed that the settings of stimulation frequency and duration varied across studies, and the effect size varied widely. The more effective conditions for PES have been studied in relation to individual parameters, such as frequency [9], pulse duration [10], and stimulation time; however, more effective conditions for combinations of multiple parameters other than the stimulus intensity and stimulus frequency have not been adequately investigated [11]. In addition, in the aforementioned clinical studies [5,6,7], although therapeutic effects were shown, various parameter settings lacked sufficient evidence from basic data. Thus, we do not have sufficient knowledge about the most effective combination of stimulation frequency and stimulation time through which PES could lead to increased excitability of the primary motor cortex for both basic and clinical studies. Therefore, in order to obtain a higher effect from PES on the improvement of motor function and to reduce the time burden of stroke patients, a more effective combination of stimulation frequency and time to increase the excitability of the primary motor cortex should be elucidated.

### Aims

Given all of this, this study aimed to explore the different combinations (stimulus duration and stimulation time) for PES to affect the excitability of the primary motor cortex in healthy participants. In this study, we examined combinations of three different stimulus frequencies (1, 10 and 50 Hz) and three different stimulus durations (20, 40 and 60 min), which were commonly used in previous studies [8,12]. This validation is expected to provide fundamental data for the development of more effective PES interventions in order to improve motor paralysis in stroke patients.

## 2. Materials and Methods

### 2.1. Participants and Experimental Procedures

The participants included 21 healthy individuals (5 men and 16 women; mean age ± standard deviation, 21.7 ± 0.6 years). The inclusion criteria were age > 18 years and right-handedness. Individuals with a history of neurological or orthopedic disorders were excluded.

Twenty-one participants experienced three different stimulus settings randomly decided in advance. The participants were seated comfortably in a reclining wheelchair in front of a table, in the prone position with their right and left forearms placed horizontally on the table and the elbow flexed to 45°. After the participants sat on the reclining wheelchair and rested for 10 min, we assessed the baseline excitability of the primary motor cortex. PES was then performed on the left upper limb at the pre-decided stimulation frequency and time, and the excitability of the primary motor cortex was evaluated after PES within 5 min. During these procedures, the participants were kept in a resting position on the reclining wheelchair. All of the participants experienced different stimulus frequency and duration combinations once a day for a total of three times, spaced 5 days apart.

### 2.2. Design of Experiments

We evaluated the more effective conditions of PES using nine combinations of stimulation frequency (1, 10 and 50 Hz) and stimulation time (20, 40 and 60 min). We used a D-optimal design of the experiment using JMP^®^ version 14.2 software (SAS Institute Inc., Cary, NC, USA). A total of 63 experimental runs proposed by the JMP^®^ software are listed in Appendix A.

### 2.3. Evaluation of the Excitability of the Primary Motor Cortex

The motor cortex excitability was evaluated using the motor-evoked potentials (MEPs) produced by TMS. Previous studies have shown alterations in the TMS-evoked MEPs following PES without concomitant changes in the brainstem electrical stimulation-evoked MEPs [3] or electrical stimulation-evoked M- and F-waves, or H-reflex [9,13,14,15], suggesting that the observed modulation occurs at the cortical level. Therefore, this study is based on the results of the above research, and experiments were conducted. TMS was delivered through a 7 cm-diameter figure-of-eight coil connected to a Magstim 200 stimulator (Magstim Co., Whitland, UK). This stimulator was placed tangentially on the scalp in an optimal position over the right hemisphere to induce maximal MEPs to the left first dorsal interosseous (FDI) muscle. The coil was placed over the primary motor cortex (M1) tangentially on the scalp, 45° away from the mid-sagittal line, approximately perpendicular to the central sulcus with the current flowing in the posterior–anterior direction.

Considering that MEP recording was performed on the resting-state muscles, the resting motor threshold (rMT) of the resting muscle was used to define the test intensity. The rMT was defined as the lowest stimulus intensity required to produce MEPs of >50 μV in at least five out of ten successive trials during the resting phase of the tested muscle [16]. The intensity of the TMS test stimulus was set at 1.2 × rMT. Ten trials were performed for each evaluation (pre-PES and post-PES).

Disposable silver–silver chloride electromyography (EMG) electrodes (1.0 cm in diameter) were placed on the left-hand FDI muscles in a belly tendon montage. The impedance was reduced to <5 kΩ. EMG signals were amplified using a conventional EMG apparatus (Power Lab 8/30, ADInstruments Pty Ltd., Bella Vista, Australia) with a band-pass filter of 2–20 kHz. Next, the signals were digitized at 4 kHz and fed into a computer for offline analysis. The EMG data were analyzed using computer software (LabChart, AD Instruments Pty Ltd., Bella Vista, Australia). The peak-to-peak amplitudes (mV) of all MEPs for the FDI muscles were calculated offline after the experiment [17]. The MEP response was expressed as the ratio to the mean value at rest before PES stimulation (MEP ratio = MEP after PES stimulation/MEP before PES stimulation).

### 2.4. Peripheral Sensory Nerve Electrical Stimulation

ESPURGE (Ito Physiotherapy and Rehabilitation, Co., Ltd., Kawaguchi, Japan) was used as the PES device. For PES, one electrode was attached to the palmar surface, approximately two-fifths distal to the palmar wrist and elbow joint of the forearm covering the ulnar nerves, and another electrode was attached 1 cm proximal to the first electrode (Figure 1). The electrical stimulation paradigm, excluding the frequency and stimulation time, was unified in the transcutaneous electrical nerve stimulation (TENS) mode with a pulse width of 1000 microseconds; the stimulation amplitude (mA) varied and was adjusted to a level above the sensory threshold but below the motor threshold, without causing discomfort. Therefore, we assumed that the electrical pulse stimulated the sensory nerve orthodromically, but not the motor nerve.

### 2.5. Statistical Analyses

JMP^®^ software version 14.2 was used for all analyses. Normal distribution was assessed using the Shapiro–Wilk test. One-way repeated-measures analysis of variance (ANOVA) was used to compare the baseline values of MEPs in the three trials. Two-way ANOVA was used to compare the baseline values of MEPs across nine PES conditions. Response surface analysis was used to identify the most effective combination within three different stimulus frequencies (1, 10 and 50 Hz) and three stimulus durations (20, 40 and 60 min). We created a prediction profile; the settings of the dialog for extracting the model were as follows: the MEP ratio was selected as the dependent variable Y, and the configuration of the model effect included frequency (Hz) and time (min) and their interaction. Individual participants were blocked and added as a mixed-effect in order to account for the effects of differences in reactivity among participants. The desirability function was used to compare and determine the most effective combination for MEP ratio among the nine different settings. The desirability value is the satisfaction index ranging between 0 and 1, where larger values are more desirable [18]. Statistical significance for all analyses was set at *p* < 0.05.

### 2.6. Ethics Approval

This study was conducted in accordance with the Declaration of Helsinki and was approved by the Ethics Committee of Shinshu University (approval no. 4698). This study was registered at the University Hospital Medical Information Network Clinical Trial Registry (registration no. UMIN000040431). All participants provided written informed consent to participate in this study.

## 3. Results

All 21 participants engaged in the experiment thrice each, for a total of 63 measurements. No adverse events were noted. There were no significant differences in the resting MEP values between the three trials for any participant (*p* = 0.292). Additionally, there were no significant differences in baseline MEP values across the nine PES conditions. The changes in the mean MEP amplitude values before and after PES are shown in Appendix A. Although differences in response to PES were observed among participants, the MEP ratio was >1.0 in 56 out of 63 trials, indicating a trend toward increased excitability of the primary motor cortex after PES. The typical recorded MEP waveforms over the left FDI muscle belonging to a representative participant are shown in Figure 2.

The summary of the responses to surface analysis is shown in Table 1. A significant effect was found for the stimulation time (LogWorth = 2.165, *p* = 0.007). However, the stimulation frequencies were not significant (LogWorth = 0.587, *p* = 0.259). Likewise, the interaction stimulation frequency × stimulation time was not significant (LogWorth = 0.601, *p* = 0.251).

The MEP changes and desirability values for each combination of stimulus frequency and duration are shown in Appendix A. According to the prediction profiler, the most effective combination of stimulation frequency and stimulation time, which maximized the value of desirability, was 10 Hz and 40 min, respectively (predicted MEP ratio: mean, 1.489; confidence interval, 1.294–1.684; value of desirability, 0.560) (Figure 3).

## 4. Discussion

We examined nine different combinations of three stimulus frequencies (1, 10 and 50 Hz) and three stimulus durations (20, 40 and 60 min) for PES in order to analyze their effects upon the excitability of the primary motor cortex in healthy participants. According to the prediction profile, the combination of stimulation frequency and stimulation time that maximized the desirability value was 10 Hz and 40 min.

Several studies have reported that PES with a stimulation frequency of 10 Hz resulted in an increased MEP [3,19,20]. Regarding the stimulation frequency, previous studies have reported that the MEP amplitude decreases at higher stimulation frequencies when compared with lower stimulation frequencies [11,21]. However, contradictory results have also been reported, with some studies reporting an increase in the MEP amplitude after PES at stimulation frequencies of >10 Hz [22,23]. In this study, no significant difference was noted in the MEP ratio across different stimulation frequency settings, and all stimulation frequency settings tended to increase the MEP ratio after PES. Therefore, the difference in stimulation frequency setting may not have a significant effect on the excitability of the primary motor cortex.

Regarding the stimulus duration, previous reviews have shown a wide diversity of findings ranging from <30 min to 2 h [8]. In this study, the MEP ratio was significantly increased when using the 40 min and 60 min settings compared to the 20 min setting. McKay et al. (2002) measured changes in MEPs at 15 min intervals for up to 2 h of neuroelectrical stimulation in healthy participants, reporting that the MEP peaked 45−60 min after the first stimulus [24]. Saito et al. also reported that there was no change in the MEP amplitude values before and after 20 min of paired-pulse electrical stimulation [25]. As the results of the present study are similar to those of previous ones, we believe that 20 min of PES stimulation is insufficient time when MEP is evaluated using the FDI muscle as the guiding muscle, and that stimulation for >40 min may increase the excitability of the primary motor cortex.

The significance of this study is that we examined more effective PES stimulation conditions from several frequency and time combinations that have not been examined extensively in previous studies. The results of this study provide useful data for the clinical application of PES. A previous review of the effects of TENS on motor function recovery in stroke patients was unable to confirm the effectiveness of this intervention owning to the heterogeneity of the stimulation protocols across studies [26]. It is necessary to verify whether the more effective settings for stimulation frequency and duration derived in this study could be applied to patients in the future. In several clinical trials with stroke patients, the stimulation frequency was set at 10 Hz [5,6,7,27,28,29]. However, the rationale for this stimulus frequency was not specified in any of these studies. In the present study, we showed that 10 Hz was not less effective than the other two conditions (1 and 50 Hz). In addition, most of the clinical studies conducted so far have applied PES for 90 min [29] to 2 h [5,6,7,27,28], which seems too long for patients. Thus, we investigated the minimum stimulation time required to reduce the time burden on patients while maintaining the intervention effectiveness. This study showed that a 40 min stimulation time was more effective than a 60 min stimulation time. Therefore, there is a possibility that the stimulation time could be reduced. However, it is also necessary to verify whether PES with a 40 min stimulation time and with a 2 h stimulation time would be equally effective.

## 5. Limitations

This study has several limitations. First, we only examined the PES effect on the excitability of the primary motor cortex using nine combinations consisting of three frequencies and three stimulation durations. Apart from these combinations, other more effective conditions (pulse width, stimulus intensity, and stimulus mode) need to be further investigated. Second, this study was conducted on healthy individuals. The participants were randomly assigned to each procedure in order to minimize the order effects of PES on MEPs. However, the fact that the differences in the participants’ responsiveness to PES may have affected the results cannot be ruled out. Finally, the FDI muscle was used as the guiding muscle. A study reported that the responsiveness to stimuli might vary across target muscles [9]. Therefore, the most effective conditions for stimulation according to different targeted nerves and muscles need to be studied.

## 6. Conclusions

We examined the effect of different combinations of stimulation frequency and durations of PES leading to the excitability of the primary motor cortex in healthy participants using the FDI muscle as the guiding muscle. Of the nine different combinations consisting of three different stimulus frequencies (1, 10 and 50 Hz) and three different stimulus durations (20, 40 and 60 min), the combination of stimulation frequency and stimulation time for increasing the MEP ratio was found to be 10 Hz and 40 min. These results were consistent with those of previous studies. It is necessary to further verify the effects of the intervention in clinical studies of stroke patients.

## Figures and Tables

**Figure 1 brainsci-12-01637-f001:**
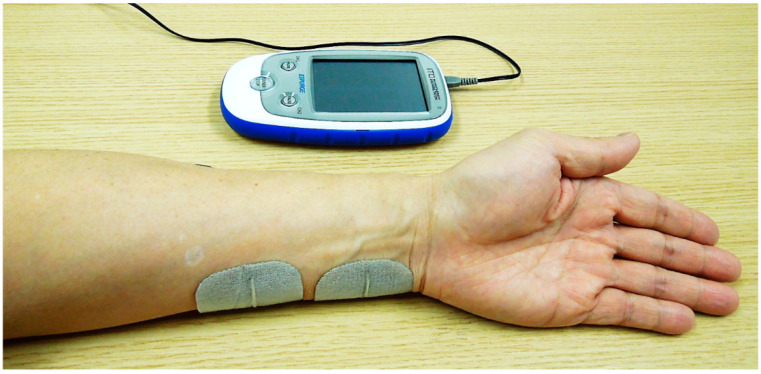
Position of the electrode attachment during electrical stimulation.

**Figure 2 brainsci-12-01637-f002:**
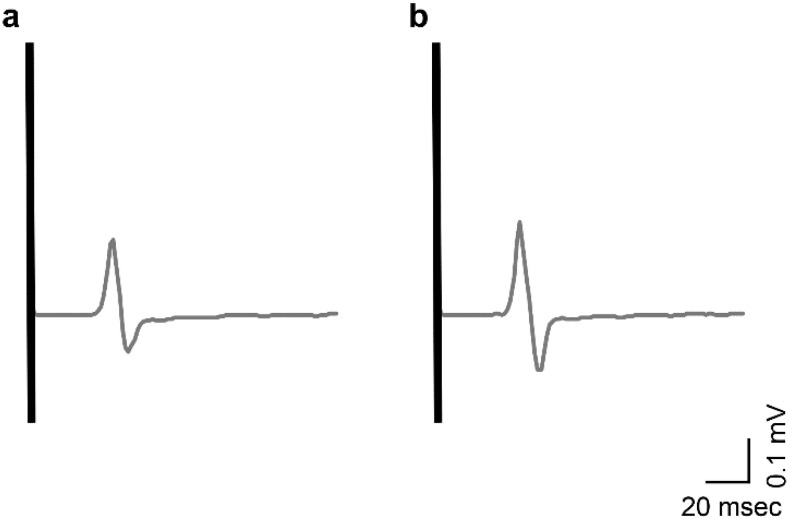
Typical MEP waveforms pre- and post-PES as recorded from a representative participant. The MEP amplitude was lower (**a**) pre-PES than (**b**) post-PES. MEP, motor-evoked potential; PES, peripheral sensory nerve electrical stimulation.

**Figure 3 brainsci-12-01637-f003:**
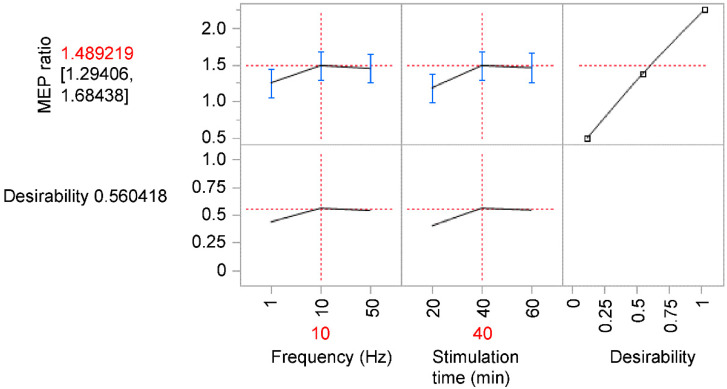
Prediction profiler produced by JMP^®^ software representing the optimum settings using desirability functions. The most effective combination of stimulation frequency and stimulation time, which maximized the value of desirability, was 10 Hz and 40 min, respectively. The error bar represents confidence interval. MEP, motor-evoked potential.

**Table 1 brainsci-12-01637-t001:** Summary of response surface.

Source	LogWorth	*p*-Value
Time (min)	2.165	0.007
Frequency (Hz) × Time (min)	0.601	0.251
Frequency (Hz)	0.587	0.259

Adjusted R^2^ = 0.31.

## Data Availability

The data used to support the findings of this study are presented in Appendix A.

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
