# Peer review of "Difference between the Effects of Peripheral Sensory Nerve Electrical Stimulation on the Excitability of the Primary Motor Cortex: Examination of the Combinations of Stimulus Frequency and Duration"

_brainsci, 2022, doi:10.3390/brainsci12121637_

Round 1
Reviewer 1 Report
This is a very interesting paper exploring the effect of different stimulus frequency and duration of peripheral sensory nerve electrical stimulation on the excitability of the primary motor cortex. The paper is well-written and of interest for the journal. However, several minor changes should be made before publishing it.
Abstract.
1-Prior to the description of the objectives and aims, I consider that a brief discussion should be made.
2-Twenty-one individuals were recruited. How were the patients selected? Please, clarify, shortly, the inclusion criteria in the abstract section.
3-The introduction should be focused on the PES in patients in stroke, as this is a conclusion in the abstract. Is the study focused in these patients? Why these findings are relevant in patients with stroke, should be introduced.
Introduction
1- The introduction is brief. I recommend to expand why this project is relevant for research in patients with stroke.
2- This study aimed to explore the different combinations of stimulus duration and time. These objectives should be expanded and clarified in a separate subsection (example. 1.1. Aims).
Methods
1- I recommend to describe first the participants and study design, and afterwards the design of experiments.
Discussion
1- The discussion is brief. I recommend to add some references about the correlation of the study results and future research on patients with neurological disorders. Stroke can be a good example.
2- A limitations and strengths section is needed. I recommend to expand it and separate it as 4.1. Limitations.
Author Response
We thank the reviewer for their thoughtful suggestions and insights and appreciate the time taken to review our manuscript and provide valuable comments. Our manuscript has benefited immensely from these insightful suggestions. I look forward to working with you to move this manuscript closer to publication in Brain Sciences.
The manuscript has been rechecked and the necessary changes have been made in accordance with the reviewers’ suggestions. We have responded to your comments as appropriate, and the responses to all comments have been prepared and attached herewith.
Thank you for your consideration. I look forward to hearing from you.
List of Responses to the Reviewer’s Comments
Reviewer #1:
This is a very interesting paper exploring the effect of different stimulus frequency and duration of peripheral sensory nerve electrical stimulation on the excitability of the primary motor cortex. The paper is well-written and of interest for the journal. However, several minor changes should be made before publishing it.
Abstract.
1-Prior to the description of the objectives and aims, I consider that a brief discussion should be made.
Response:
Thank you for the suggestion. We have added the following text prior to the description of the objectives and aims.
Page 1, lines 13–16.
Peripheral sensory nerve electrical stimulation (PES) excites the primary motor cortex and is expected to improve motor dysfunction post-stroke. However, previous studies have reported a variety of stimulus frequencies and stimulus duration settings, and the effects of these different combinations on primary motor cortex excitability are not clear.
In addition, we have modified or deleted several sentences so that the total word count for the entire abstract is no more than 200 words.
2-Twenty-one individuals were recruited. How were the patients selected? Please, clarify, shortly, the inclusion criteria in the abstract section.
Response:
Thank you for the suggestion. Accordingly, we have added the following sentence to specify the inclusion criteria for the participants.
Page 1, lines 18–19.
Twenty-one healthy individuals (aged >18 years, right-handedness, without a history of neurological or orthopedic disorders) were included.
3-The introduction should be focused on the PES in patients in stroke, as this is a conclusion in the abstract. Is the study focused in these patients? Why these findings are relevant in patients with stroke, should be introduced.
Response:
Thank you for your valuable comment. As pointed out by the reviewer, we have added the introduction to describe the focus of our study on PES for stroke patients.
Page 1, lines 13–16.
Peripheral sensory nerve electrical stimulation (PES) excites the primary motor cortex and is expected to improve motor dysfunction post-stroke. However, previous studies have reported a variety of stimulus frequencies and stimulus duration settings, and the effects of these different combinations on primary motor cortex excitability are not clear.
Introduction
1- The introduction is brief. I recommend to expand why this project is relevant for research in patients with stroke.
Response:
Thank you for your valuable comment. We have added several sentences to expand the Introduction, particularly on the relevance of this research for patients with stroke.
Pages 1–2, lines 41–44.
Activation of the primary motor cortex by PES is thought to be effective in improving motor paralysis in stroke patients, and the effects of interventions combining various rehabilitation approaches and PES have recently been reported [5–7].
Page 2, lines 55–57.
In addition, in the aforementioned clinical studies [5-7], although therapeutic effects were shown, various parameter settings lacked sufficient evidence from basic data.
Page 2, lines 57–63.
Thus, we do not have sufficient knowledge about the more effective combination of stimulation frequency and stimulation time through which PES could lead to increased excitability of the primary motor cortex for both basic and clinical studies. Therefore, to obtain a higher effect of PES on the improvement of motor function and reduce the time burden of stroke patients, a more effective combination of stimulation frequency and time to increase the excitability of the primary motor cortex should be elucidated.
2- This study aimed to explore the different combinations of stimulus duration and time. These objectives should be expanded and clarified in a separate subsection (example. 1.1. Aims).
Response:
Thank you for the suggestion. As suggested by the reviewer, we have added a separate subsection in 1.Introduction.
Page 2, lines 64
1.1. Aims
In addition, we have added the following text to clarify the purpose of this study.
Page 2, 69–71
This validation is expected to provide fundamental data for the development of more effective PES interventions to improve motor paralysis in stroke patients.
Methods
1- I recommend to describe first the participants and study design, and afterwards the design of experiments.
Response:
Thank you for the suggestion. Accordingly, we have changed the order of the description in the Methods section
Pages 2–3, lines 73–95.
2.1. Participants and Experimental Procedures
2.2. Design of Experiments
Discussion
1- The discussion is brief. I recommend to add some references about the correlation of the study results and future research on patients with neurological disorders. Stroke can be a good example.
Response:
Thank you for your valuable comment. We have added several sentences and references to describe the correlation of the study results and future research on patients with stroke.
Page 6, lines 227–238
In several clinical trials with stroke patients, the stimulation frequency was set at 10 Hz [5-7,27-29]. However, the rationale for this stimulus frequency was not specified in any of these studies. In the present study, we showed that 10 Hz was not less effective than the other two conditions (1 and 50 Hz). In addition, most of the clinical studies conducted so far have applied PES for 90 min [29] to 2h [5-7,27,28], which seems too long for patients. Thus, we investigated the minimum stimulation time required to reduce the time burden on patients while maintaining the intervention effectiveness. This study showed that a 40-min stimulation time was more effective than a 60-min stimulation time. Therefore, there is a possibility that the stimulation time could be reduced. However, it is also necessary to verify whether PES with a 40-min stimulation time and with a 2-h stimulation time would be equally effective.
【Added references】
- Sawaki L, Wu CW, Kaelin-Lang A, Cohen LG. Effects of somatosensory stimulation on use-dependent plasticity in chronic stroke. Stroke. 2006 Jan;37(1):246-7. doi: 10.1161/01.STR.0000195130.16843.ac.
- Menezes IS, Cohen LG, Mello EA, Machado AG, Peckham PH, Anjos SM, Siqueira IL, Conti J, Plow EB, Conforto AB. Combined Brain and Peripheral Nerve Stimulation in Chronic Stroke Patients With Moderate to Severe Motor Impairment. Neuro-modulation. 2018 Feb;21(2):176-183. doi: 10.1111/ner.12717.
- Conforto AB, Machado AG, Menezes I, Ribeiro NHV, Luccas R, Pires DS, Leite CDC, Plow EB, Cohen LG. Treatment of Upper Limb Paresis With Repetitive Peripheral Nerve Sensory Stimulation and Motor Training: Study Protocol for a Randomized Controlled Trial. Front Neurol. 2020 Mar 25;11:196. doi: 10.3389/fneur.2020.00196.
2- A limitations and strengths section is needed. I recommend to expand it and separate it as 4.1. Limitations.
Response:
Thank you for the suggestion. We have added the separate subsection in 4.Discussion.
Page 6, lines 239
4.1. Limitations
In addition, we have corrected several sentences in the Abstract and Main Text to ensure adequate and clear explanation.

Reviewer 2 Report
The authors performed extensive and exciting research, and stimulation parameters based on scientific evidence are needed in practice.
I only have one suggestion.
The authors should carefully verify the TENS machine specifications. I doubt the pulse duration was 1 ms since TENS is delivered in a pulse width of microseconds.
Author Response
We thank the reviewer for their thoughtful suggestions and insights and appreciate the time taken to review our manuscript and provide valuable comments. Our manuscript has benefited immensely from these insightful suggestions. I look forward to working with you to move this manuscript closer to publication in Brain Sciences.
The manuscript has been rechecked and the necessary changes have been made in accordance with the reviewers’ suggestions. We have responded to your comments as appropriate, and the responses to all comments have been prepared and attached herewith.
Thank you for your consideration. I look forward to hearing from you.
List of Responses to the Reviewer’s Comments
Reviewer #2:
The authors performed extensive and exciting research, and stimulation parameters based on scientific evidence are needed in practice.
I only have one suggestion.
The authors should carefully verify the TENS machine specifications. I doubt the pulse duration was 1 ms since TENS is delivered in a pulse width of microseconds.
Response:
Thank you for your valuable suggestion. We have corrected the unit for pulse duration from 1 ms to 1,000 microseconds.
Page 3, lines 131–133
The electrical stimulation paradigm, excluding the frequency and stimulation time, was unified in the transcutaneous electrical nerve stimulation (TENS) mode with a pulse width of 1,000 microseconds,
In addition, we have corrected several sentences in the Abstract and Main Text to ensure adequate and clear explanation.
